# Dynamic Modelling to Describe the Effect of Plant Extracts and Customised Starter Culture on *Staphylococcus aureus* Survival in Goat’s Raw Milk Soft Cheese

**DOI:** 10.3390/foods12142683

**Published:** 2023-07-12

**Authors:** Beatriz Nunes Silva, Sara Coelho-Fernandes, José António Teixeira, Vasco Cadavez, Ursula Gonzales-Barron

**Affiliations:** 1Centro de Investigação de Montanha (CIMO), Instituto Politécnico de Bragança, Campus de Santa Apolónia, 5300-253 Bragança, Portugal; beatrizsilva@ceb.uminho.pt (B.N.S.); sara.coelho@ipb.pt (S.C.-F.); vcadavez@ipb.pt (V.C.); 2Laboratório para a Sustentabilidade e Tecnologia em Regiões de Montanha, Instituto Politécnico de Bragança, Campus de Santa Apolónia, 5300-253 Bragança, Portugal; 3CEB—Centre of Biological Engineering, University of Minho, 4710-057 Braga, Portugal; jateixeira@deb.uminho.pt; 4LABBELS—Associate Laboratory, 4710-057 Braga, Portugal

**Keywords:** predictive microbiology, lactic acid bacteria, antagonism, dairy, artisanal cheese, pH, lemon balm, sage, spearmint

## Abstract

This study characterises the effect of a customised starter culture (CSC) and plant extracts (lemon balm, sage, and spearmint) on *Staphylococcus aureus* (SA) and lactic acid bacteria (LAB) kinetics in goat’s raw milk soft cheeses. Raw milk cheeses were produced with and without the CSC and plant extracts, and analysed for pH, SA, and LAB counts throughout ripening. The pH change over maturation was described by an empirical decay function. To assess the effect of each bio-preservative on SA, dynamic Bigelow-type models were adjusted, while their effect on LAB was evaluated by classical Huang models and dynamic Huang–Cardinal models. The models showed that the bio-preservatives decreased the time necessary for a one-log reduction but generally affected the cheese pH drop and SA decay rates (logD_ref_ = 0.621–1.190 days; controls: 0.796–0.996 days). Spearmint and sage extracts affected the LAB specific growth rate (0.503 and 1.749 ln CFU/g day^−1^; corresponding controls: 1.421 and 0.806 ln CFU/g day^−1^), while lemon balm showed no impact (*p* > 0.05). The Huang–Cardinal models uncovered different optimum specific growth rates of indigenous LAB (1.560–1.705 ln CFU/g day^−1^) and LAB of cheeses with CSC (0.979–1.198 ln CFU/g day^−1^). The models produced validate the potential of the tested bio-preservatives to reduce SA, while identifying the impact of such strategies on the fermentation process.

## 1. Introduction

The occurrence of *Staphylococcus aureus* in milk and cheeses has been documented several times [1,2,3,4,5,6], and multiple dairy-related outbreaks have been linked to this pathogen [7,8,9,10]. Its occurrence is generally associated with subclinical or clinical mastitis in dairy cattle, which contaminates the milk [11], and with the lack of appropriate hygienic measures during cheese production, as *S. aureus* can be detected on the hair, nostrils, skin, pharynx, and mucosa of humans, as well as in the gastrointestinal and urogenital tracts [12], even though water, milking equipment, and the environment are other sources of contamination [1]. Contamination of milk with *S. aureus* is particularly relevant in the case of cheeses produced with raw milk, since there is no pathogen inactivation step, such as pasteurisation, although *S. aureus* may also be found in pasteurised milk cheeses if unhygienic practices lead to the recontamination of thermally treated milk [12].

Various plant extracts have been added to cheeses and other dairy products as bio-preservative agents, considering their antimicrobial capacities. Mohamed et al. [13] successfully tested the use of *Moringa oleifera* leaves’ extract as an antimicrobial agent in cream cheese against several pathogens, Carvalho et al. [14] used *Thymus mastichina* extracts to inhibit *Staphylococcus* spp. and *Enterobacteriaceae* in raw milk cheeses, and Shan et al. [15] investigated the antimicrobial efficiency of clove, oregano, cinnamon stick, grape seed extracts, and pomegranate peel against *Staphylococcus aureus*, *Listeria monocytogenes*, and *Salmonella* enterica in cheese, reporting that all five extracts were active against the pathogens.

Intentionally added lactic acid bacteria (LAB) with known antimicrobial activity may be another strategy to enhance the safety of milk and cheeses and has also been previously tested by other researchers. For example, Gonzales-Barron et al. [16] observed a significant decline of *L. monocytogenes* throughout the ripening of artisanal Minas semi-hard cheese when adding a combination of LAB strains with anti-listerial capacity, Le Marc et al. [17] validated a commercial starter culture (Fresco 1010, Chr. Hansen, Hørsholm, Denmark) as successful in inhibiting the growth of *S. aureus* during milk fermentation, and Alomar et al. [18] co-cultured, separately, *Enterococcus* and *Lactococcus* strains with *S. aureus*, and the pathogen growth was inhibited after 6 h of incubation in microfiltered milk.

Nevertheless, only the use of LAB that belong to the genera *Lactococcus*, *Lactobacillus*, *Leuconostoc*, *Pediococcus*, and some *Streptococcus* is approved by the European Food Safety Authority (EFSA), which established its Qualified Presumption of Safety (QPS) status that indicates that there is reasonable evidence that such microorganisms do not raise safety concerns [19]. The *Enterococcus* genus and some *Streptococcus* species, however, can be pathogenic and present resistance to various antibiotics and virulence factors, and for that reason, they do not have QPS status [19].

Previous research from our laboratory [20,21] has shown, through determination of their minimum inhibitory concentration (MIC), the in vitro antibacterial activity of lemon balm, spearmint, and sage extracts against *S. aureus*, thus suggesting their potential to be included in foods and act against microbial spoilage. More specifically, hydroethanolic (70% (*v*/*v*)) lemon balm extract obtained by solid–liquid extraction had a MIC of 2.5 mg/mL against *S. aureus*, while the same extracts obtained from sage and spearmint showed a MIC of 0.625 mg/mL and 1.25 mg/mL against this pathogen, respectively [20]. A customised starter culture is also suggested in this work as a result of previous investigation from our research team [22]. After collecting LAB isolates (N = 232) from cheeses produced with goat’s raw milk, determining their antibacterial, acidifying, and proteolytic activities, and conducting molecular characterisation, four strains were selected (*Leuconostoc mesenteroides*, *Lacticaseibacillus paracasei*, *Lactococcus cremoris*, and *Lactococcus lactis*) to compose a cocktail of LAB that could be implemented as a starter culture in cheese manufacture, considering its bactericidal and acidogenic capacities.

Mathematical models can be used for microbial behaviour predictions, as the determination of growth parameters of pathogens can be used to assess and manage the risk of foodborne illnesses [23]. In this sense, one of the objectives of this work was to mathematically determine the effect of a customised starter culture and of lemon balm, sage, and spearmint extracts (when directly incorporated in curd, during cheese production) against *S. aureus* in goat’s raw milk cheeses, and to characterise *S. aureus* survival kinetics through a Bigelow model. Using this approach, the decimal reduction times (*D*) can be described as a function of pH and incorporation of starter culture or plant extract, and the survival parameters may support the optimisation of the cheese-making process to improve cheeses’ microbial safety. Another objective was to investigate the impact of the plant extracts and starter culture on the evolution of LAB, to ensure that this microbial community, crucial for the fermentation process, is not negatively affected, and to compare the behaviour of autochthonous LAB with that of LAB when a starter culture is added.

## 2. Materials and Methods

### 2.1. Plant Materials’ and Extracts’ Preparation

Dried aerial parts of lemon balm, spearmint, and sage were supplied by Pragmático Aroma Lda. (“Mais Ervas”, Trás-os-Montes, Portugal) and mechanically ground. Extracts were produced according to the methodology of Silva et al. [20]. Briefly, the extractions were performed with ethanol 70% (*v*/*v*) as a solvent in a water bath, with agitation (150 rpm) at 60 °C for 90 min. The ratio between the sample and the solvent was 1:20. After filtrating the mixtures (7–10 μm), the ethanolic fraction was evaporated, whereas the aqueous fraction was frozen and lyophilised.

### 2.2. Bacterial Strains

*Staphylococcus aureus* ATCC 6538, from the Polytechnic Institute of Bragança stock collection, was used. A loop of culture maintained on Nutrient Agar slant was cultivated two times at 37 °C, for 16 h, at 200 rpm, first on tryptic soy broth (TSB) and then on TSB with pH adjusted to 6.34, to mimic goat’s milk pH. On the day of cheese production, centrifugation (10,640× *g* at 4 °C for 10 min) of the second culture was performed to remove debris and culture media, and after the supernatant was discarded, pellets were washed with sterile 0.9% physiological solution. These steps were performed twice, and cells were resuspended in sterile 0.9% physiological solution to achieve approximately 7 log CFU/mL.

For the LAB cocktail, four strains of LAB (*Leuconostoc mesenteroides*, *Lacticaseibacillus paracasei*, *Lactococcus cremoris*, and *Lactococcus lactis*), isolated from goats’ raw milk artisanal cheeses and that presented antimicrobial and acidifying activity [22], were used in this work. For the preparation of individual LAB strain suspensions, the cryopreserved strains were thawed, and a loop of culture of each strain was separately cultivated at 30 °C for 24 h in MRS broth. Two successive inoculations were then performed by placing 100 μL of the subcultures in 10 mL of MRS broth at 30 °C for 24 h. The following inoculation was carried out by placing 500 μL of the subculture in 200 mL of MRS broth at 30 °C for 18 h, to achieve a concentration of each strain of approximately 9 log CFU/mL, adjusted by measuring absorbance at 600 nm using a spectrophotometer (Peak Instruments Inc., Version 1701, Houston, TX, USA). Equal parts of each strain were then combined to obtain the selected LAB cocktail.

### 2.3. Inoculation of Bacterial Strains in Milk and Cheese Production: Incorporation of Starter Culture or Plant Extract

To prepare laboratory-scale cheeses, rennet (0.75 mL/L milk) and *S. aureus* inoculum (5 mL/L milk) were added to milk at approximately 34 °C, in the case of challenge tests with plant extracts, or rennet (0.75 mL/L milk), *S. aureus* inoculum (5 mL/L milk), and selected LAB cocktail inoculum (10 mL/L milk, 1% (*v*/*v*)), in the case of challenge tests with added starter culture. With this approach, each cheese contained around 4 to 5 log CFU/g of *S. aureus*, depending on the initial milk contamination.

After 30 min at 34 °C, curdled milk was diced and drained, and for challenge tests with plant extracts, 1% (*w*/*w*) of lyophilised sage, lemon balm, or spearmint extract was incorporated into the curd and mixed. An inoculated control without extract or without starter culture was kept. Non-inoculated cheeses with starter culture were also produced.

Next, the curd was transferred into 50 mL tubes which were subjected to centrifugation at 6000 rpm, 20 °C, for 3.5 min. The supernatant, i.e., whey, was discarded, and the compacted curd was cut into cheeses of approximately 5 g, which were then placed in a 15% (*w*/*v*) brine solution (cheese:brine ratio of approximately 90 g:1.5 L) for 10 min at 25 °C for salting. Lastly, the weight (g) of each cheese was registered and cheeses were stored in a climate-controlled chamber (10 °C, 98% RH) for 15 days for fermentation and maturation to occur.

### 2.4. Microbiological and Physicochemical Analysis throughout Cheese Ripening

Analyses were performed between day 0 (day of cheese manufacture) and day 15. For every test unit, for the microbiological analysis, suitable serial dilutions were prepared after homogenisation of the cheese in buffered peptone water (50 mL) for 60 s. The *S. aureus* concentration was determined by plating a 0.1 mL aliquot on Baird–Parker agar supplemented with Egg Yolk Tellurite, according to the ISO norm [24]. Typical colonies were counted after incubation at 37 °C for 48 h.

The LAB concentration was determined by incorporating a 1 mL aliquot in MRS agar (for plant extracts challenge tests) or MRS and M17 agar (for the starter culture challenge test), and overlaying with 1.2% bacteriological agar, following the ISO norm [25]. Then, enumeration of typical colonies was performed, following incubation of plates for 48 h at 30 °C.

Physicochemical analyses during cheese ripening comprised pH and water activity measurements. The first was carried out using a pH meter (Hanna Instruments, model HI5522, Smithfield, RI, USA) with a HI1131 glass penetration probe. For the latter, samples were placed in an Aqualab meter cuvette (4TE Decagon, Pullman, DC, USA), and the value was annotated after measurement stabilisation.

### 2.5. Modelling of S. aureus and LAB Behaviour during Cheese Ripening

#### 2.5.1. *S. aureus* Behaviour during Cheese Ripening

Since the pH of cheese varied during ripening at 10 °C, and changes in *S. aureus* counts were primarily driven by the drop in pH, dynamic kinetic analysis was employed to assess the *S. aureus* kinetic parameters in cheese.

For every treatment, a three-parameter empirical decay function was used to describe the pH change over the maturation time, as follows:(1)$${pH}_{t}=\left({pH}_{0}-{pH}_{res}\right)\times e^{-k_{pH}t}+{pH}_{res}$$
where *k_pH_* is the pH decay rate (day^−1^), ${pH}_{t}$ is the pH at time *t*, ${pH}_{0}$ is the pH at time 0, and ${pH}_{res}$ is the asymptotic pH.

Then, a differential log-decay function with shoulder and tail as the primary model (as described by Geeraerd et al. [26,27]), with a changing *D*-value, linked to a Bigelow equation of the *D*-value as a function of pH (with parameters $\log D_{ref}$ at pH 7.0 and $z_{pH}$) as the secondary model, was adjusted:$$\frac{dN}{dt}=-kN\left(\frac{1}{1+C_{c}}\right)\left(1-\frac{N_{res}}{N}\right)$$
(2)$$\frac{dC_{c}}{dt}=-kC_{c}$$
$$D=\frac{ln(10)}{k}$$
(3)$$\log D=\log D_{ref}-{\left(\frac{pH-{pH}_{ref}}{z_{pH}}\right)}^{2}$$

In Equation (2), *N* is the population density (CFU/g); *k* is the decay rate of *S. aureus* (day^−1^); $C_{c}$ is related to the damage level caused to the population, as its value influences the decay rate, k, and therefore allows for the existence or absence of a shoulder region in the survival curve; and $N_{res}$ is the residual population density and is related to the tailing phenomenon in the survival curve (CFU/g). In Equation (3), *D* is the decimal reduction time (day) at 10 °C and at the cheese pH, ${pH}_{ref}$ is the reference pH (fixed to 7.0), $z_{pH}$ is the difference of pH from ${pH}_{ref}$ that causes a ten-fold change in decimal reduction time, and $D_{ref}$ is the decimal reduction time at ${pH}_{ref}$ (days). First, $C_{c}$ and $N_{res}$ were approximated by fitting Equation (2) to each dataset in integrated form, and subsequently, $z_{pH}$ and $\log D_{ref}$ were estimated by fitting the dynamic model defined by Equations (2) and (3).

#### 2.5.2. LAB Behaviour during Cheese Ripening

**Cheeses with plant extracts.** To evaluate the impact of plant extracts on the growth of LAB in cheese, the integrated Huang model [28], as described in Equation (4), was used:$$Y\left(t\right)=Y_{0}+Y_{max}-ln(e^{Y_{0}}+(e^{Y_{max}}-e^{Y_{0}})e^{-\mu_{max}\beta (t)})$$
(4)$$\beta \left(t\right)=t+\frac{1}{\alpha }ln\left(\frac{1+e^{-\alpha (t-\lambda )}}{1+e^{\alpha \lambda }}\right)$$

In Equation (4), $Y_{0}$ and $Y_{max}$ are the natural logarithms of the initial (time 0) and maximum microbial concentrations, respectively, and Y is the natural logarithm of the bacterial concentration at the “real time” *t* (ln CFU/g). $\mu_{max}$ is the maximum specific growth rate (ln CFU/g day^−1^), β(t) is the transition function, λ is the lag phase (day) of the growth curve, α is the lag phase transition coefficient (dimensionless), and *t* is the time (day) under a fixed temperature (10 °C). The parameter α was set to 4.0, as suggested by Huang [29]. The estimated parameters from Equation (4) were $Y_{0}$, $Y_{max}$, and $\mu_{max}$. It is acknowledged that the changing pH of the matrix affects the growth rate of LAB in cheese; nonetheless, a dynamic model was not fitted to the LAB data since the objective was to compare the effect of the extracts on LAB growth, and not to characterise the kinetic parameters of the pool of indigenous lactic acid bacteria, which were largely unknown at that time.

**Cheeses with the selected LAB cocktail.** To calculate the kinetic parameters of LAB, taking into consideration the pH decay throughout storage at 10 °C (constant temperature), dynamic kinetic analysis was used. This was carried out by fitting, at once, a primary growth model in differential form with a secondary model of the specific growth rate as affected by the cheese pH. The Huang model [28] was selected as the primary model to describe the growth of LAB in cheese during ripening, and the cardinal parameter model for pH was picked for the secondary modelling [30]. Accordingly, the following model, labelled as Huang–Cardinal, was fitted to the data:$$\frac{dY}{dt}=\frac{\mu_{max}}{1+e^{-\alpha \left(t-\lambda \right)}}\left(1-e^{Y-Y_{max}}\right)$$
(5)$$\mu_{max}=\mu_{opt}\left\{\frac{\left(pH-{pH}_{min}\right)(pH-{pH}_{max})}{\left(pH-{pH}_{min}\right)\left(pH-{pH}_{max}\right)-{\left(pH-{pH}_{opt}\right)}^{2}}\right\}$$

As in Equation (4), in Equation (5), $Y_{max}$ and Y represent the maximum level of the bacterial concentration and the bacterial concentration at time *t*, respectively (ln CFU/g), $\mu_{max}$ is the maximum specific growth rate (ln CFU/g day^−1^), λ is the lag phase duration (day) of the growth curve (set to zero), α is the lag phase transition coefficient (dimensionless, set to 4.0 [29]), and *t* is the time (day) under 10 °C. ${pH}_{min}$ and ${pH}_{max}$ refer to the minimum and maximum pH values that allow microbial growth, respectively, and ${pH}_{opt}$ is the pH at which the $\mu_{max}$ is optimal.

The estimated parameters from Equation (5) were $Y_{0}$, $Y_{max}$, and $\mu_{opt}$. The latter parameter represents the optimum specific growth rate of LAB in goats’ raw milk soft cheese at 10 °C and at the optimum pH for growth (${pH}_{opt}$), considering the water activity variation during ripening as negligible. The cardinal parameters of LAB (${pH}_{min}$, ${pH}_{opt}$, and ${pH}_{max}$) were not estimable from our data due to the small range of the monitored pH of cheeses (5.04–6.61). For that reason, literature data [31,32,33,34,35,36,37,38,39,40,41,42,43,44,45] were used to set the average cardinal values of the LAB strains used in the cocktail (*L. mesenteroides*, *L. paracasei*, *L. cremoris*, and *L. lactis*) as: ${pH}_{min}$ = 4.00, ${pH}_{opt}$ = 6.50, and ${pH}_{max}$ = 9.00.

#### 2.5.3. Estimation of Parameters

Ordinary differential equations such as Equations (2) and (5), which do not have an analytical solution, can be solved using numerical methods. Numerical optimisation implies seeking for the model parameters resulting in least residual sum of squares of the errors. Herein, the fourth-order Runge–Kutta method was used to resolve the ordinary differential equations [28], whereas the unknown kinetic parameters were determined by least-square optimisation, employing the libraries ‘deSolve’ and ‘FME’ from the R software (version 4.1.0, R Foundation for Statistical Computing, Vienna, Austria). The mean absolute error (MAE, Equation (6)) and root mean square error (RMSE, Equation (7)) were also calculated to evaluate the models’ fitting capacities, as:(6)$$MAE=\frac{\sum \left|Y_{obs\;i}-Y_{fit\;i}\right|}{n}$$
(7)$$RMSE=\sqrt{\frac{\sum {\left(Y_{obs\;i}-Y_{fit\;i}\right)}^{2}}{df}}$$
where $Y_{fit\;i}$ and $Y_{obs\;i}$ represent each of the *i*-th *S. aureus* or LAB concentrations adjusted by the model and their corresponding observations, respectively. The degrees of freedom (df) were calculated as ‘n-np’, with n being the number of data points of a growth curve and np being the number of parameters of the model.

## 3. Results and Discussion

### 3.1. pH Decay during Cheese Ripening

The change in pH throughout cheese ripening is shown in Figure 1 (plant extracts) and Figure 2 (selected LAB cocktail). The estimated parameters of the empirical decay function used to describe the pH change over the maturation time are presented in Table 1.

The natural course of the fermentation process appeared to be impacted by the presence of spearmint and lemon balm extracts, as suggested by the lower pH drop rates, *k_pH_*, of the treatments (spearmint: 0.194 day^−1^; lemon balm: 0.223 day^−1^), in comparison to the corresponding controls (spearmint: 0.262 day^−1^; lemon balm: 0.240 day^−1^), whereas sage extract had no effect on the pH drop rate (0.521 day^−1^ and 0.522 day^−1^ for cheeses with and without sage extract, respectively). In this sense, among the extracts tested, spearmint affected the pH drop rate the most.

However, sage extract also affected the fermentation process, but in this case it was the extract with the biggest impact on the final pH achieved: cheeses with this extract presented a greater difference between their ${pH}_{res}$ (5.377) and the ${pH}_{res}$ of the control treatment (5.172), in comparison to cheeses with and without spearmint (5.584 and 5.418, respectively) and with and without lemon balm extracts (5.286 and 5.115, respectively). Nevertheless, in all cases, the ${pH}_{res}$ was higher in cheeses with plant extracts, compared to the controls.

The pH drop rate in *S. aureus*-free cheeses with the addition of the selected LAB cocktail was lower ($k_{pH}$ = 0.263 day^−1^; Figure 2, bottom plot) than that of cheeses inoculated with *S. aureus* (without LAB cocktail: $k_{pH}$ = 0.330 day^−1^; with LAB cocktail: $k_{pH}$ = 0.337 day^−1^; Figure 2, top plots). However, the pH of *S. aureus*-free cheeses with the selected LAB cocktail by the end of the challenge test was much lower (${pH}_{res}$ = 5.250) when compared to that of cheeses inoculated with *S. aureus*, either with (${pH}_{res}$ = 5.656) or without (${pH}_{res}$ = 5.576) the selected LAB cocktail. While the strains composing the customised starter culture presented an acidifying capacity in vitro [22], it seems that, in this challenge test, they were not able to accelerate the pH decay during fermentation, as would have been expected [46]. In any case, the selected LAB cocktail promoted a decay more prolonged in time, which enabled reaching a lower ${pH}_{res}$ by the end of maturation.

Comparing cheeses inoculated with *S. aureus* but not the selected LAB cocktail (Figure 2, top left plot) with those inoculated with both (Figure 2, top right plot), it can be seen that the starter culture only slightly modified the pH drop rate (without the LAB cocktail: $k_{pH}$ = 0.330 day^−1^; with the LAB cocktail: $k_{pH}$ = 0.337 day^−1^), and that cheeses inoculated with *S. aureus* and the selected LAB cocktail were not able to reach a pH value as low as those inoculated with *S. aureus* only (${pH}_{res}$ = 5.656 vs. ${pH}_{res}$ = 5.576).

Changes in water activity were observed during cheese ripening, with values oscillating between 0.932 and 0.984 without a specific trend, so no function could be fitted to describe the water activity evolution over time.

### 3.2. S. aureus Behaviour during Cheese Ripening

*S. aureus* survival, as affected by lemon balm, spearmint, or sage extracts, or the selection of LAB, was described by Bigelow-type secondary models.

The pathogen’s survival curves in cheese with plant extracts and with a selected LAB cocktail, as portrayed by dynamic models, are displayed in Figure 3. The dynamic survival model was not fitted for the treatment without the selected LAB cocktail as *S. aureus* decay did not occur. The results of this particular control (slight growth of *S. aureus*) were, therefore, not quite aligned with those of the control treatments for the extracts (survival of *S. aureus*). Since the experimental work conducted was the same for the controls of all runs, the normal lot-to-lot variations in the composition, microbiota, and microbiological quality of the raw milk are likely to explain the slight deviation observed in such control treatment.

Table 2 shows the outcomes of the Bigelow parameters for each treatment.

The models suitably fitted the survival curves, with RMSE values of 0.116, 0.063, 0.057, and 0.103 for spearmint, lemon balm, sage, and the selected LAB cocktail, respectively, producing significant estimates in all cases (*p* < 0.05).

From Table 2, $\log D_{ref}$ was influenced by the use of extracts (0.621 ± 0.061 days for spearmint; 1.190 ± 0.200 for lemon balm; 0.996 ± 0.278 for sage), in comparison to the controls (0.993 ± 0.190 days for spearmint; 0.996 ± 0.056 for lemon balm; 0.796 ± 0.068 for sage).

In cheeses with spearmint extract, $\log D_{ref}$ was smaller than that of the control (0.621 ± 0.061 vs. 0.993 ± 0.190 days), thus suggesting a superior decay rate of *S. aureus*. Moreover, the survival curves in Figure 3 show that the addition of spearmint extract decreased the shoulder and promoted *S. aureus* decline earlier in maturation. In turn, comparing the survival curves of the treatments without and with the selected LAB cocktail, it seems that the customised starter culture completely inverted the behaviour of *S. aureus*, as it inhibited the pathogens’ growth observed in the control and started promoting *S. aureus* decay after around five days of ripening.

On the other hand, when incorporating lemon balm or sage extract into the cheese, $\log D_{ref}$ was higher (1.190 ± 0.200 for lemon balm; 0.996 ± 0.278 for sage) than that of the controls (0.996 ± 0.056 for lemon balm; 0.796 ± 0.068 for sage), which implies a lower decline rate. Nonetheless, when these extracts were incorporated in cheese, *S. aureus* decay was more stable and extended throughout ripening, in comparison to control cheeses, in which the *S. aureus* decay phase was brief, and the stationary phase (tail) was reached faster (Figure 3).

The use of plant extracts reduced the time needed to reach a one-log decrease of *S. aureus*, which in practical terms was shown by a reduction of 0.634 log CFU/g (sage), 0.611 log CFU/g (lemon balm), and 1.373 log CFU/g (spearmint) after 12 days of maturation (Table 2). Without the incorporation of plant extracts, *S. aureus* decay was still observed but less pronounced, with a decline in the pathogen concentration between 0.238 and 0.491 log CFU/g in the same period. Considering these results, the effectiveness of using spearmint, lemon balm, and sage extracts to reduce *S. aureus* in raw milk cheeses was confirmed. The addition of the selected LAB cocktail also reduced the time necessary for a log decrease, and in practice corresponded to a reduction of 0.493 log CFU/g after 12 days of maturation. These results are coherent with previous works reporting on the antimicrobial capacities of selected LAB strains [16,47,48,49,50,51] and plant extracts [13,14,15,52,53] against various microorganisms in cheeses.

The higher $z_{pH}$ of cheeses with extract of spearmint (3.172 ± 0.655) and extract of lemon balm (2.340 ± 0.835) in Table 2 indicate that a bigger difference between pH and ${pH}_{ref}\;$ is required for a ten-fold change in *D* when adding these bio-preservatives to cheese than the one necessary for the same change in *D* in the controls (spearmint: 1.599 ± 0.358; lemon balm: 1.851 ± 0.066). This would imply that, for the same pH variation, *S. aureus* in cheeses with incorporated spearmint or lemon balm extract would suffer a smaller reduction than *S. aureus* in control cheeses; however, a phenomenon of interaction should also be considered in the interpretation, since the addition of extracts to the curd retarded the pH drop (Table 1). Other inhibitory mechanisms apart from pH decay may promote pathogen decline, and in the mathematical equations, these could manifest themselves in the shortening of the shoulders. On the other hand, the $z_{pH}$ value of cheeses with sage extract (2.006 ± 0.677) was close to that of the control (2.054 ± 0.131), suggesting that the difference between pH and ${pH}_{ref}$ that leads to a ten-fold change in *D* is virtually the same in both cases.

Overall, the results of the Bigelow-type secondary models indicate that the plant extracts and selected starter culture tested may be used for the control of *S. aureus* in cheeses, but that each bio-preservative influences different factors. The results showed that the main effect of introducing 1% lemon balm extract or 1% sage extract in curd was on the delay of the *S. aureus* stationary phase and the $z_{pH}$ parameter, whereas 1% spearmint extract affected the *S. aureus* shoulder, $z_{pH}$, and $\log D_{ref}$. In turn, the main effect of the selected LAB cocktail was on $z_{pH}$ and $\log D_{ref}$, as it inverted the behaviour of *S. aureus* from growth to survival.

Considering the multiple impacts on the pathogen and the reduction promoted, among all options, spearmint extract appears to be more effective in controlling *S. aureus* in cheeses made with goat’s raw milk. This was despite the previously determined MIC of spearmint against *S. aureus*, which was not the lowest among the three extracts tested (lemon balm extract: 2.5 mg/mL; spearmint: 1.25 mg/mL; sage: 0.625 mg/mL) [20], thus demonstrating the effect of the matrix on the antimicrobial potential of bio-preservatives [54].

From our previous work [20], spearmint extract did not present the highest total phenolic content but revealed the highest concentration of rosmarinic acid (333 mg/L extract) when compared to sage (170 mg/L extract) and lemon balm extracts (185 mg/L extract). Rosmarinic acid is recognised for its high antimicrobial capacity [55,56], and although the mechanisms of action are not fully known, Honório et al. [57] reported cell shrinkage and the occurrence of blebbing-like formations on *S. aureus* cell surfaces, and Bais et al. [58] also described damaged cell surfaces when treating *A. niger* with rosmarinic acid. Ferulic acid, ellagic acid, naringin, hesperidin, resveratrol, and quercetin were also detected in our spearmint hydroethanolic extract, but in lower concentrations compared to rosmarinic acid [20]. The antimicrobial potential of these compounds against *S. aureus* has also been reported in the literature [59,60,61,62,63,64].

### 3.3. LAB Behaviour during Cheese Ripening


**Cheeses with plant extracts.**


The Huang model parameters describing the behaviour of LAB in goat’s raw milk cheese with and without plant extracts during maturation are shown in Table 3. The corresponding fitted models are depicted in Figure 4.

The Huang model adequately fitted each of the growth curves, with RMSE between 0.275 and 0.654, and produced significant parameter estimates (*p* < 0.05).

In the case of inoculated cheeses produced with sage extract, the growth curve of LAB presented a lag phase (λ = 1.749 ± 0.565 days; Figure 4), which did not happen in non-inoculated cheeses with sage. This suggests that the combined presence of *S. aureus* and sage extract acts as a hurdle against LAB, inducing a period of adaptation before cell growth is possible. The other extracts tested did not produce this response.

The estimated initial LAB concentration, $Y_{0}$, varied between different experiments (from 13.94 ± 0.366 to 18.27 ± 0.338 ln CFU/g), a consequence of the high microbial variability of the raw milk used for cheese production.

In inoculated cheeses, significant differences (*p* < 0.05) were detected between the initial concentration of LAB in cheeses without and with lemon balm (17.31 ± 0.168 and 16.71 ± 0.346, respectively), and also without and with spearmint extracts (14.11 ± 0.360 and 15.00 ± 0.299 ln CFU/g day^−1^, correspondingly), although no differences were found in $Y_{0}$ between cheeses produced with and without sage extracts (18.27 ± 0.338 and 18.19 ± 0.325 ln CFU/g day^−1^, respectively).

Regarding the $\mu_{max}$ parameter, in inoculated cheeses, the specific growth rate of LAB was not affected by the incorporation of lemon balm extract, as reflected by the estimated values of 0.960 ± 0.038 and 0.967 ± 0.073 ln CFU/g day^−1^ (0.417 and 0.420 log CFU/g day^−1^; *p* > 0.05) and the identical shape of the growth curves in Figure 4. On the other hand, spearmint and sage extracts considerably modified (*p* < 0.05) the exponential phase of LAB in cheeses with *S. aureus* (observe the distinct growth curve shapes in Figure 4). Spearmint incorporation triggered a lower specific growth rate of LAB (0.503 ± 0.076 compared to 1.421 ± 0.189 ln CFU/g day^−1^ for cheeses without extract, or 0.218 and 0.617 log CFU/g day^−1^, correspondingly), whereas sage reduced the cell-doubling time, i.e., increased the specific growth rate (1.749 ± 0.565 vs. 0.806 ± 0.102 ln CFU/g day^−1^ for cheeses without extract, or 0.760 vs. 0.350 log CFU/g day^−1^).

In the case of cheeses with lemon balm and spearmint extracts, the negative impact of the presence of *S. aureus* on the LAB specific growth rate was observable, as significant differences (*p* < 0.05) were found between $\mu_{max}$ of inoculated and non-inoculated cheeses (the latter being higher). However, in cheeses produced with sage extract, the opposite was observed, as inoculated cheeses revealed higher $\mu_{max}$, 1.133 ± 0.174 ln CFU/g day^−1^ (0.492 log CFU/g day^−1^), than those non-inoculated, 0.643 ± 0.138 ln CFU/g day^−1^ (0.279 log CFU/g day^−1^). Regardless of the direction of change, differences in $\mu_{max}$ between the two treatments may be partly explained by microbial competition mechanisms between LAB and *S. aureus*.

The extracts did not have an impact on the maximum LAB concentration, as no significant differences were detected between the $Y_{max}$ values of cheeses with and without either of the plant extracts (in inoculated samples). However, the presence of *S. aureus* in cheeses with lemon balm and sage extracts appears to influence $Y_{max}$, as visible in the plots of Figure 4: non-inoculated cheeses with lemon balm reached a higher LAB final concentration (27.79 ± 0.349 ln CFU/g), whereas in the case of cheeses with sage extract, inoculated samples were the ones achieving greater $Y_{max}$ values (24.05 ± 0.229 ln CFU/g).

Even though cheeses with lower $Y_{0}$ presented lower $Y_{max}$ values (spearmint 0% and 1%), it could not be inferred that the maximum concentration achieved is influenced by the initial LAB numbers, since treatments with higher $Y_{0}$ (sage 0% and 1%) did not present the highest $Y_{max}$.

Considering these results, lemon balm extract appears to be the one affecting LAB behaviour the least. In turn, spearmint extract greatly reduced (by more than half) the specific growth rate of LAB, even though, by the end of maturation, the same concentration was achieved. Taking into account the results in Table 2, where the high antagonistic effect of this extract against *S. aureus* is observed, it seems that spearmint extract exhibits high antimicrobial capacity against both microbial communities. Sage extract and *S. aureus* contamination caused a period of little to no cell division (lag phase), but the higher specific growth rate allowed the cells to reach the stationary phase earlier, when comparing inoculated cheeses with and without sage, with no impact on the final LAB concentration reached.

**Cheeses with the selected LAB cocktail.** The Huang–Cardinal model parameters describing the behaviour of LAB in goat’s raw milk cheese with and without a cocktail of the selected LAB during maturation are shown in Table 4 and Table 5, for LAB isolated in MRS agar and M17 agar, respectively.

All models properly fitted the growth curves, with RMSE values between 0.120 and 0.248, and produced significant estimates (*p* < 0.05). Nonetheless, it is noteworthy to point out that the estimates of the kinetic parameters of M17-grown LAB (Table 5) were associated with higher standard errors than those isolated in MRS agar (Table 4). This is a consequence of the lower selectivity of M17 agar compared to MRS agar, which causes higher variability in the results of the microbiological analysis (plate counting) and, therefore, affects the precision of the estimation of parameters.

From Table 4, the Huang–Cardinal models revealed that adding the selected LAB with antimicrobial activity reduced the $\mu_{opt}$ (1.198 ± 0.260 and 1.144 ± 0.091 ln CFU/g day^−1^, or 0.520 and 0.497 log CFU/g day^−1^) and increased the $Y_{max}$ (20.22 ± 0.199 and 20.40 ± 0.071 ln CFU/g) of MRS-grown LAB in comparison to cheeses without the addition of the selected LAB cocktail ($\mu_{opt}$: 1.560 ± 0.260 and 1.343 ± 0.145 ln CFU/g day^−1^, or 0.677 and 0.583 log CFU/g day^−1^, and $Y_{max}$: 18.54 ± 0.137 and 18.83 ± 0.085 ln CFU/g). The estimates in Table 5 suggest the same tendencies in cheeses with intentionally added LAB: a reduction of the optimum specific growth rate, $\mu_{opt}$ (0.979 ± 0.236 and 1.372 ± 0.246 ln CFU/g day^−1^, or 0.425 and 0.596 log CFU/g day^−1^), and an increase of the $Y_{max}$ (21.20 ± 0.265 and 20.88 ± 0.236 ln CFU/g). In addition to the anticipated increase in the $Y_{max}$, the initial concentration, $Y_{0}$, was also higher in cheeses with incorporation of the selected LAB (MRS: 16.37 ± 0.144 and 16.65 ± 0.340 ln CFU/g; M17: 16.72 ± 0.464 and 15.99 ± 0.392 ln CFU/g).

Overall, the results of the Huang–Cardinal models indicate that, regardless of the absence or presence of *S. aureus*, at 10 °C and pH = 6.50 (assumed as optimum), autochthonous LAB grew at a higher rate than those present in cheeses with the addition of the selected LAB cocktail, although they did not reach such high final concentrations. Previous work by Cadavez et al. [23] observed the same trend in terms of a reduction of the LAB growth rate, since treatments without the selected anti-listerial LAB cocktail presented greater LAB growth rates than those with the addition of the customised starter, as estimated by Jameson-effect models. Gonzales-Barron et al. [16] and Campagnollo et al. [47] also observed higher growth rates of native LAB in comparison to the growth rates of LAB in treatments with a selected and deliberately added starter culture. In these studies, the authors pointed out that the lower growth rate of LAB in cheeses with the addition of a selected LAB cocktail could be a consequence of the initial LAB concentration, $Y_{0}$, being higher, and therefore, closer to the maximum carrying capacity, and/or a result of intra-species competition between native LAB and intentionally added LAB [16,23]. These explanations also apply to our study.

The influence of the pH on the specific growth rate of LAB can be appreciated in Figure 5, which shows values predicted by the underlying cardinal model. The plots illustrate that cheese acidification during ripening caused a decrease of the specific growth rate of LAB for all cheese treatments, i.e., the pH evolution of cheese was towards the lower limit for bacterial growth. To this end, as maturation elapses, the lower pH values may directly affect the cells or cause an increase of the degree of dissociation of organic acids [65], thus reducing the growth potential of the LAB.

## 4. Conclusions

The Bigelow-type secondary models characterised *S. aureus* survival parameters in goat’s raw milk cheese produced with plant extracts (lemon balm, sage, spearmint) or with a customised LAB starter culture during cold maturation, and were able to confirm and quantitatively describe the inhibitory effect of the selected plant extracts and the selected LAB cocktail on *S. aureus*.

The results of the Bigelow-type secondary models showed that both $\log D_{ref}$ and $z_{pH}$ were influenced by the addition of extracts and the use of the starter culture. $z_{pH}$ values increased with the addition of extracts as a compensatory effect of the slower pH drop caused by these bio-preservatives. The dynamic models also revealed that the implementation of any of the bio-preservation strategies tested reduced the time required to reduce *S. aureus* by one log, thus supporting their capability to act as antimicrobial agents during cheese maturation.

The Huang models pointed to lemon balm extract as the one affecting LAB behaviour the least, whereas spearmint extract greatly reduced the specific growth rate of LAB, although the same final concentration was achieved as that of the control. In turn, the results of the Huang–Cardinal model revealed that autochthonous LAB grew at a higher rate than those of cheeses with a cocktail of the selected LAB, and this was independent of the inoculation of *S. aureus*.

The models developed in this work validated the bio-preservatives tested as adequate strategies to reduce *S. aureus* contamination and improve the safety of raw milk cheeses. Furthermore, the results also pointed to the effects of such preservatives on the fermentation parameters. The importance of monitoring the pH decay of cheeses during maturation when incorporating plant extracts was evidenced. In case the appropriate pH drop during fermentation is compromised by the addition of herbal extracts and this affects the quality of the final product, it may be necessary to investigate and implement a solution to overcome this hindrance. Further challenge tests may be directed towards investigating the combined effects of using a starter culture with a high acidification capacity and adding herbal extracts of proven inhibitory effects against *S. aureus*. Lastly, it is important to remark that the models in this work do not account for the effect of temperature, so they cannot be employed to estimate kinetic parameters at temperatures other than 10 °C.

## Figures and Tables

**Figure 1 foods-12-02683-f001:**
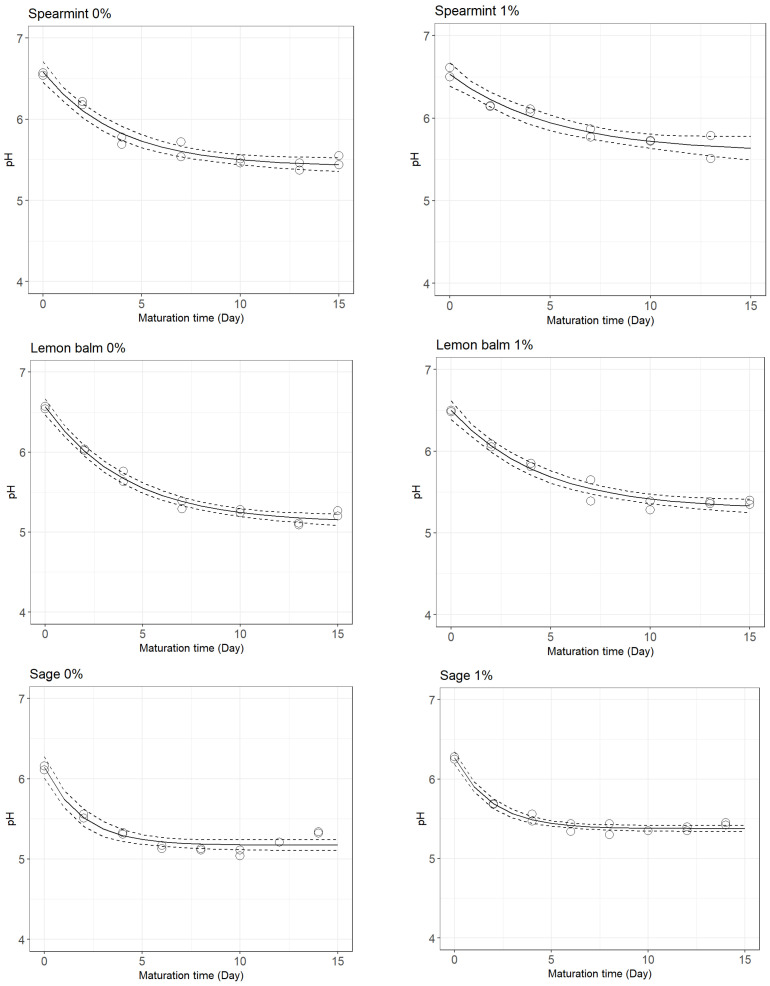
Change in pH of lab-scale cheeses made from goats’ raw milk inoculated with *S. aureus* without (**left**) and with (**right**) the addition of 1% (*w*/*w*) spearmint, lemon balm, and sage extracts, as described by a three-parameter empirical decay function (full line) with 95% confidence intervals, CI (dashed lines).

**Figure 2 foods-12-02683-f002:**
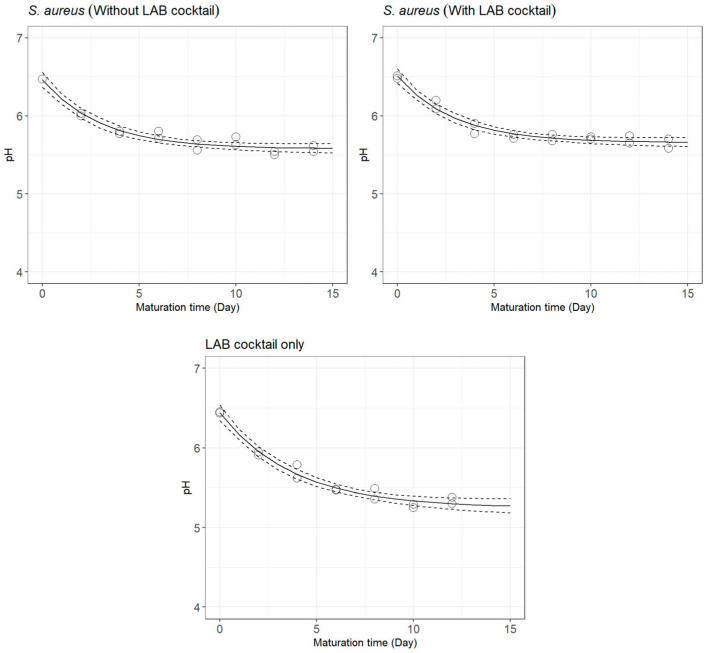
Change in pH of lab-scale cheeses made from goats’ raw milk inoculated with *S. aureus* without (**top left**) and with (**top right**) the addition of a cocktail of selected LAB, and cheeses non-inoculated with *S. aureus* with the selected LAB cocktail (**bottom**), as described by a three-parameter empirical decay function (full line) with the 95% CI (dashed lines).

**Figure 3 foods-12-02683-f003:**
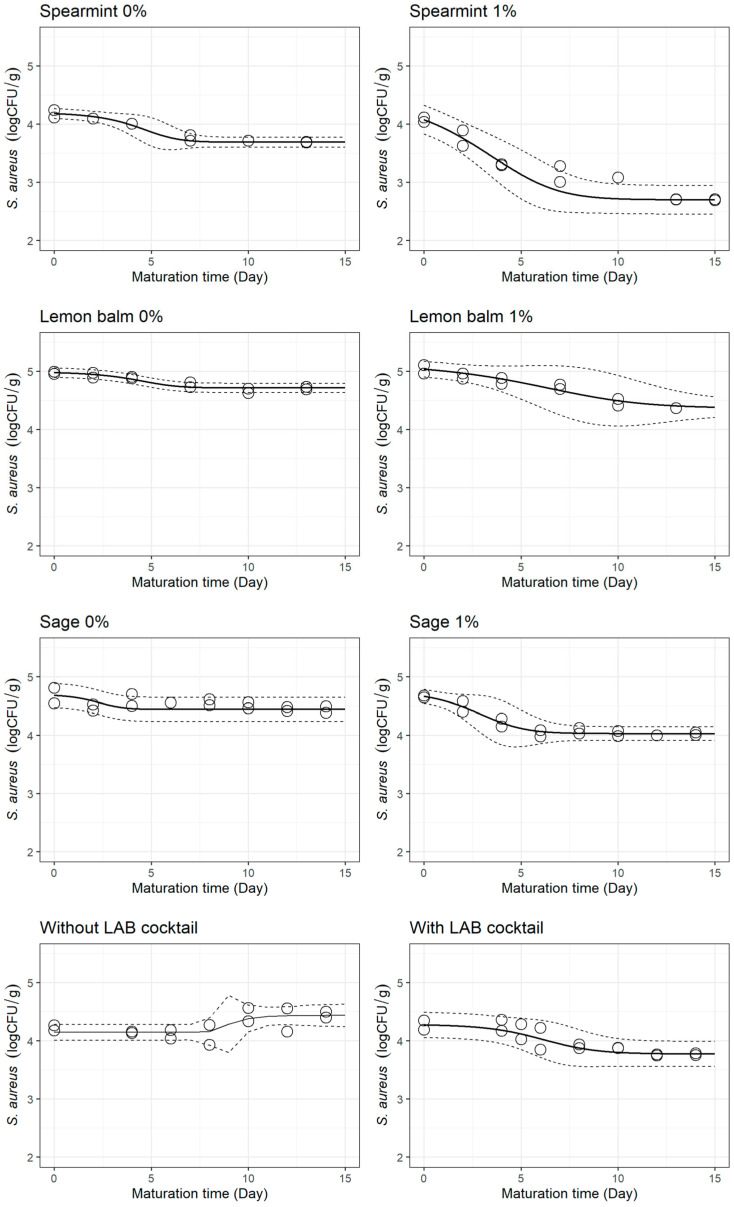
*S. aureus* behaviour in goat’s raw milk cheese without or with 1% (*w*/*w*) of extract of spearmint, lemon balm, or sage, and with a cocktail of the selected LAB as a starter culture, as depicted by dynamic modelling (full lines) with the 95% CI (dashed lines). As an exception, the integrated Huang model was fitted to the curve produced without the LAB cocktail.

**Figure 4 foods-12-02683-f004:**
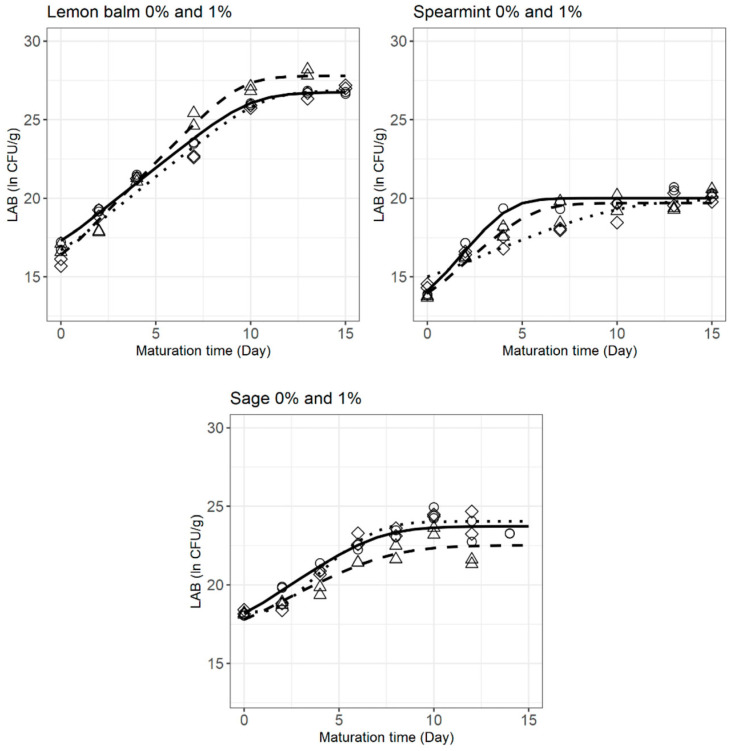
Growth of LAB in goat’s raw milk cheese inoculated with *S. aureus*, with (⋯·◊⋯·) and without (^__^○^__^) plant extracts, and non-inoculated with plant extracts (- -∆- -), as depicted by the Huang model. The same markers represent observations from the same experiment.

**Figure 5 foods-12-02683-f005:**
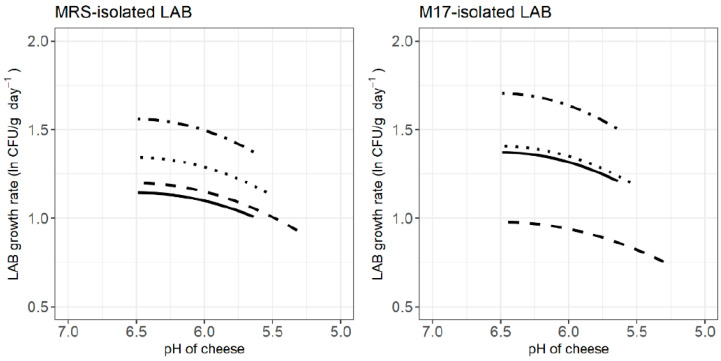
Effect of pH on the specific growth rate (ln CFU/g day^−1^) of MRS-isolated (**left**) and M17-isolated LAB (**right**) in goat’s raw milk cheese inoculated with *S. aureus*, with (**^___^**) and without (**^…^**) the selected LAB cocktail, and non-inoculated, with (- - -) and without the selected LAB cocktail (-•-), as shown by the Huang–Cardinal parameter model.

**Table 1 foods-12-02683-t001:** Effect of the addition of spearmint, lemon balm, or sage extract in curd or of a cocktail of selected LAB on the parameters of the empirical decay function used to describe the pH change over the maturation time in goat’s raw milk cheese, goodness-of-fit measures (S^2^, RMSE, and MAE), and the estimated pH decay (*pH*_0_−*pH_res_*) throughout maturation.

Treatment	Parameters	Mean ± SE	Pr (>|t|)	Goodness-of-Fit Measures	${pH}_{0}-{pH}_{res}$
*S. aureus* + Spearmint 0%	${pH}_{0}$	6.581 ± 0.058	<0.0001	S^2^ = 0.006 RMSE = 0.075 MAE = 0.061	1.163
	${pH}_{res}$	5.418 ± 0.050	<0.0001		
	$k_{pH}$	0.262 ± 0.041	<0.0001		
*S. aureus* + Spearmint 1%	${pH}_{0}$	6.530 ± 0.062	<0.0001	S^2^ = 0.007 RMSE = 0.079 MAE = 0.067	0.946
	${pH}_{res}$	5.584 ± 0.107	<0.0001		
	$k_{pH}$	0.194 ± 0.058	0.008		
*S. aureus* + Lemon balm 0%	${pH}_{0}$	6.567 ± 0.046	<0.0001	S^2^ = 0.004 RMSE = 0.059 MAE = 0.047	1.452
	${pH}_{res}$	5.115 ± 0.043	<0.0001		
	$k_{pH}$	0.240 ± 0.025	<0.0001		
*S. aureus* + Lemon balm 1%	${pH}_{0}$	6.502 ± 0.053	<0.0001	S^2^ = 0.005 RMSE = 0.069 MAE = 0.050	1.216
	${pH}_{res}$	5.286 ± 0.055	<0.0001		
	$k_{pH}$	0.223 ± 0.034	<0.0001		
*S. aureus* + Sage 0%	${pH}_{0}$	6.142 ± 0.061	<0.0001	S^2^ = 0.007 RMSE = 0.079 MAE = 0.063	0.970
	${pH}_{res}$	5.172 ± 0.031	<0.0001		
	$k_{pH}$	0.522 ± 0.092	<0.0001		
*S. aureus* + Sage 1%	${pH}_{0}$	6.265 ± 0.036	<0.0001	S^2^ = 0.002 RMSE = 0.046 MAE = 0.037	0.888
	${pH}_{res}$	5.377 ± 0.018	<0.0001		
	$k_{pH}$	0.521 ± 0.058	<0.0001		
*S. aureus* without LAB cocktail	${pH}_{0}$	6.461 ± 0.044	<0.0001	S^2^ = 0.003 RMSE = 0.057 MAE = 0.045	0.885
	${pH}_{res}$	5.576 ± 0.031	<0.0001		
	$k_{pH}$	0.330 ± 0.047	<0.0001		
*S. aureus* with LAB cocktail	${pH}_{0}$	6.509 ± 0.042	<0.0001	S^2^ = 0.003 RMSE = 0.054 MAE = 0.043	0.853
	${pH}_{res}$	5.656 ± 0.029	<0.0001		
	$k_{pH}$	0.337 ± 0.047	<0.0001		
LAB cocktail only	${pH}_{0}$	6.440 ± 0.044	<0.0001	S^2^ = 0.004 RMSE = 0.058 MAE = 0.044	1.190
	${pH}_{res}$	5.250 ± 0.051	<0.0001		
	$k_{pH}$	0.263 ± 0.035	<0.0001		

**Table 2 foods-12-02683-t002:** Effect of the addition of spearmint, lemon balm, or sage extract in curd or of a cocktail of the selected LAB on the Bigelow parameters of *S. aureus* in goat’s raw milk cheese along maturation, goodness-of-fit measures (S^2^, RMSE, and MAE), and *S. aureus* mean decay (log CFU/g) after 12 days.

Treatment	Parameters	Mean ± SE	Pr (>|t|)	Goodness-of-Fit Measures	△Y_0–12_ (log CFU/g)
Spearmint 0% ($C_{c}$(0) = 1.5)	log D_ref_	0.993 ± 0.190	0.001	S^2^ = 0.002 RMSE = 0.040 MAE = 0.035	0.491
	z_pH_	1.599 ± 0.358	<0.0001		
Spearmint 1% ($C_{c}$(0) = 0.01)	log D_ref_	0.621 ± 0.061	<0.0001	S^2^ = 0.015 RMSE = 0.116 MAE = 0.098	1.373
	z_pH_	3.172 ± 0.655	<0.0001		
Lemon balm 0% ($C_{c}$(0) = 1.5)	log D_ref_	0.996 ± 0.056	<0.0001	S^2^ = 0.002 RMSE = 0.037 MAE = 0.033	0.262
	z_pH_	1.851 ± 0.066	<0.0001		
Lemon balm 1% ($C_{c}$(0) = 0.01)	log D_ref_	1.190 ± 0.200	<0.0001	S^2^ = 0.004 RMSE = 0.063 MAE = 0.056	0.611
	z_pH_	2.340 ± 0.835	0.019		
Sage 0% ($C_{c}$(0) = 1.5)	log D_ref_	0.796 ± 0.068	<0.0001	S^2^ = 0.010 RMSE = 0.098 MAE = 0.077	0.238
	z_pH_	2.054 ± 0.131	<0.0001		
Sage 1% ($C_{c}$(0) = 0.01)	log D_ref_	0.996 ± 0.278	0.003	S^2^ = 0.003 RMSE = 0.057 MAE = 0.047	0.634
	z_pH_	2.006 ± 0.677	0.010		
With LAB cocktail ($C_{c}$(0) = 3)	log D_ref_	0.756 ± 0.067	<0.0001	S^2^ = 0.011 RMSE = 0.103 MAE = 0.078	0.493
	z_pH_	2.490 ± 0.487	<0.0001		

**Table 3 foods-12-02683-t003:** Kinetic parameters (initial and maximum microbial concentration, Y_0_ and Y_max_, in ln CFU/g, maximum specific growth rate, $\mu_{max}$, in ln CFU/g day^−1^, and lag duration, λ, in days) of LAB in goat’s raw milk cheese subjected to various treatments during maturation, as estimated by the Huang model, and goodness-of-fit measures (S^2^, RMSE, and MAE).

Treatment	Parameters	Mean ± SE	Pr (>|t|)	Goodness-of-Fit Measures
Spearmint 1%	$Y_{0}$	13.94 ± 0.366	<0.0001	S^2^ = 0.277 RMSE = 0.508 MAE = 0.445
	$Y_{max}$	19.70 ± 0.216	<0.0001	
	$\mu_{max}$	1.088 ± 0.160	<0.0001	
*S. aureus* + Spearmint 0%	$Y_{0}$	14.11 ± 0.360	<0.0001	S^2^ = 0.254 RMSE = 0.486 MAE = 0.438
	$Y_{max}$	20.01 ± 0.195	<0.0001	
	$\mu_{max}$	1.421 ± 0.189	<0.0001	
*S. aureus* + Spearmint 1%	$Y_{0}$	15.00 ± 0.299	<0.0001	S^2^ = 0.249 RMSE = 0.481 MAE = 0.421
	$Y_{max}$	19.99 ± 0.358	<0.0001	
	$\mu_{max}$	0.503 ± 0.076	<0.0001	
Lemon balm 1%	$Y_{0}$	16.40 ± 0.307	<0.0001	S^2^ = 0.252 RMSE = 0.275 MAE = 0.208
	$Y_{max}$	27.79 ± 0.349	<0.0001	
	$\mu_{max}$	1.219 ± 0.074	<0.0001	
*S. aureus* + Lemon balm 0%	$Y_{0}$	17.31 ± 0.168	<0.0001	S^2^ = 0.081 RMSE = 0.275 MAE = 0.208
	$Y_{max}$	26.75 ± 0.157	<0.0001	
	$\mu_{max}$	0.960 ± 0.038	<0.0001	
*S. aureus* + Lemon balm 1%	$Y_{0}$	16.71 ± 0.346	<0.0001	S^2^ = 0.361 RMSE = 0.579 MAE = 0.490
	$Y_{max}$	26.87 ± 0.350	<0.0001	
	$\mu_{max}$	0.967 ± 0.073	<0.0001	
Sage 1%	$Y_{0}$	17.80 ± 0.435	<0.0001	S^2^ = 0.461 RMSE = 0.654 MAE = 0.539
	$Y_{max}$	22.52 ± 0.457	<0.0001	
	$\mu_{max}$	0.643 ± 0.138	<0.0001	
*S. aureus* + Sage 0%	$Y_{0}$	18.19 ± 0.325	<0.0001	S^2^ = 0.262 RMSE = 0.496 MAE = 0.369
	$Y_{max}$	23.73 ± 0.232	<0.0001	
	$\mu_{max}$	0.806 ± 0.102	<0.0001	
*S. aureus* + Sage 1%	$Y_{0}$	18.27 ± 0.338	<0.0001	S^2^ = 0.179 RMSE = 0.408 MAE = 0.336
	$Y_{max}$	24.05 ± 0.229	<0.0001	
	$\mu_{max}$	1.133 ± 0.174	<0.0001	
	λ	1.749 ± 0.565	0.011	

**Table 4 foods-12-02683-t004:** Kinetic parameters (initial and maximum microbial concentrations, Y_0_ and Y_max_, in ln CFU/g, and optimum specific growth rate, $\mu_{opt}$, in ln CFU/g day^−1^) of MRS-isolated LAB in goat’s raw milk cheese subjected to various treatments during maturation, as estimated by the Huang–Cardinal model, and goodness-of-fit measures (S^2^, RMSE, and MAE).

Treatment	Parameters	Mean ± SE	Pr (>|t|)	Goodness-of-Fit Measures
Without LAB cocktail	$Y_{0}$	15.01 ± 0.258	<0.0001	S^2^ = 0.040 RMSE = 0.183 MAE = 0.135
	$Y_{max}$	18.54 ± 0.137	<0.0001	
	$\mu_{opt}$	1.560 ± 0.260	0.009	
With LAB cocktail	$Y_{0}$	16.65 ± 0.340	<0.0001	S^2^ = 0.074 RMSE = 0.248 MAE = 0.198
	$Y_{max}$	20.22 ± 0.199	<0.0001	
	$\mu_{opt}$	1.198 ± 0.260	0.019	
*S. aureus* without LAB cocktail	$Y_{0}$	15.16 ± 0.174	<0.0001	S^2^ = 0.021 RMSE = 0.134 MAE = 0.120
	$Y_{max}$	18.83 ± 0.085	<0.0001	
	$\mu_{opt}$	1.343 ± 0.145	0.0007	
*S. aureus* with LAB cocktail	$Y_{0}$	16.37 ± 0.144	<0.0001	S^2^ = 0.016 RMSE = 0.120 MAE = 0.096
	$Y_{max}$	20.40 ± 0.071	<0.0001	
	$\mu_{opt}$	1.144 ± 0.091	<0.0001	

**Table 5 foods-12-02683-t005:** Kinetic parameters (initial and maximum microbial concentrations, Y_0_ and Y_max_, in ln CFU/g, and optimum specific growth rate, $\mu_{opt}$, in ln CFU/g day^−1^) of M17-isolated LAB in goat’s raw milk cheese subjected to various treatments during maturation, as estimated by the Huang–Cardinal model, and goodness-of-fit measures (S^2^, RMSE, and MAE).

Treatment	Parameters	Mean ± SE	Pr (>|t|)	Goodness-of-Fit Measures
Without LAB cocktail	$Y_{0}$	15.27 ± 0.531	<0.0001	S^2^ = 0.190 RMSE = 0.404 MAE = 0.349
	$Y_{max}$	19.57 ± 0.249	<0.0001	
	$\mu_{opt}$	1.705 ± 0.475	0.023	
With LAB cocktail	$Y_{0}$	16.72 ± 0.464	<0.0001	S^2^ = 0.168 RMSE = 0.3791 MAE = 0.342
	$Y_{max}$	21.20 ± 0.265	<0.0001	
	$\mu_{opt}$	0.979 ± 0.236	0.014	
*S. aureus* without LAB cocktail	$Y_{0}$	15.35 ± 0.321	<0.0001	S^2^ = 0.074 RMSE = 0.253 MAE = 0.232
	$Y_{max}$	19.96 ± 0.164	<0.0001	
	$\mu_{opt}$	1.407 ± 0.221	0.003	
*S. aureus* with LAB cocktail	$Y_{0}$	15.99 ± 0.392	<0.0001	S^2^ = 0.102 RMSE = 0.292 MAE = 0.261
	$Y_{max}$	20.88 ± 0.236	<0.0001	
	$\mu_{opt}$	1.372 ± 0.246	0.011	

## Data Availability

Summary data are available upon request.

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
