# Peer review of "Dynamic Modelling to Describe the Effect of Plant Extracts and Customised Starter Culture on Staphylococcus aureus Survival in Goat’s Raw Milk Soft Cheese"

_foods, 2023, doi:10.3390/foods12142683_

Round 1
Reviewer 1 Report
1. μmax should be the specific growth rate with the unit of ln CFU/g/[time], or commonly abbreviated as [time]-1. Frequently, the unit of the growth rate is log10 CFU/g/[time], with is 2.303 (ln(10)) times smaller than the specific growth rate. Thus, please check the descriptions in L25, L199, L203, L220, L226, and so on.
2. I would suggest using different symbols to express the pH decay (drop?) rate and the specific inactivation rate of microorganisms.
3. Please ensure that all measurements, such as pH, counts, and growth rates, are consistently formatted and properly reported.
L42 …in the case of/that cheeses…
L47 successfully
L82 cocktail
Reviewer 2 Report
The manuscript deals with a quantitative description of the mutual effects between various plant extracts and LAB and S. aureus populations that are generally in competition during goat´s cheese maturation. To reach sound results that were able to evaluate and compare to each other, the authors used well-chosen mathematical growth and inactivation/survival models (as well as the combination of primary and secondary models). The work was mainly orientated towards plant extracts and LAB behaviour. Confirmation of the decrease of S. aureus numbers was also provided using Bigelow-type secondary models with the outputs (log Dref and zpH) well-established in practise.
The topic of the manuscript is relevant as the safety of artisanal cheeses is still discussed, as well as the fate of the microbial populations during cheese ripening or the effects of some added plant extracts (that could be accepted in goat´s cheeses) on both LAB or undesirable contaminants.
Allow me to share with the authors the following comments:
To better understand the numeric values of growth rates, I would expect to use decimal logs.
Using a cocktail of four LAB - three species of cocci, one of the lactobacilli - that were used in cultivation in MRS broth. This could "deform" the intended ratio of LAB before inoculation, since MRS medium is more suitable for lactobacilli than lactococci. Generally, lactococci are more needed to start fermentation and grow well at an early stage of cheese ripening.
From the growth modelling of LAB, the high initial numbers of inoculum could not provide sufficient space for them to grow and reliably describe the effects of observed factors. The LAB growth potential was only about 5 to 6 ln CFU/g (appr. 2,3 log10). However, the presented statistical indices showed low and acceptable values, including errors in all the model parameters.
The conclusions are consistent and with relevant outputs for cheese practise.
Additional remarks to the authors:
Please, unify the unit of growth rates in the manuscript (day-1 vs. days-1; L256-260). Better to use the simple abbreviation of d-1.
Add a better explanation of the meaning of Cc in lines 182-183. The original source is more specific referring to this parameter.
L310, 355: Is it really about inactivation? Use it carefully within the context. It needs to be reviewed and checked through the manuscript.
Figure 4. There are no clear differences among the figures.
Table 4: Too complicated head of the table - “... isolated in MRS agar in goat’s raw milk cheese during maturation with and without a cocktail of selected LAB and inoculated or not with S. aureus..."
Generally, in my opinion, commenting only on the outputs in natural logarithms does not seem to be practical. Sometimes, the decimal logs could be used in brackets.
